# Web-Based Application for Biomedical Image Registry, Analysis, and Translation (BiRAT)

Rahul Pemmaraju [1], Robert Minahan [2], Elise Wang [3], Kornel Schadl [4], Heike Daldrup-Link [5] and Frezghi Habte [5,*]

1   School of Bioengineering and Biomedical Engineering, Johns Hopkins University, Baltimore, MD 21218, USA; rpemmar1@jhu.edu
2   Computational and Systems Biology, University of California-Los Angeles, Los Angeles, CA 90095, USA; rminahan@ucla.edu
3   School of Medicine, University of Rochester, Rochester, NY 14642, USA; elise_wang@urmc.rochester.edu
4   Department of Orthopedic Surgery, Stanford School of Medicine, Stanford, CA 94305, USA; kornels@stanford.edu
5   Department of Radiology, Stanford School of Medicine, Stanford, CA 94305, USA; heiked@stanford.edu
*   Correspondence: fhabte@stanford.edu

**Abstract:** Imaging has become an invaluable tool in preclinical research for its capability to non-invasively detect and monitor disease and assess treatment response. With the increased use of preclinical imaging, large volumes of image data are being generated requiring critical data management tools. Due to proprietary issues and continuous technology development, preclinical images, unlike DICOM-based images, are often stored in an unstructured data file in company-specific proprietary formats. This limits the available DICOM-based image management database to be effectively used for preclinical applications. A centralized image registry and management tool is essential for advances in preclinical imaging research. Specifically, such tools may have a high impact in generating large image datasets for the evolving artificial intelligence applications and performing retrospective analyses of previously acquired images. In this study, a web-based server application is developed to address some of these issues. The application is designed to reflect the actual experimentation workflow maintaining detailed records of both individual images and experimental data relevant to specific studies and/or projects. The application also includes a web-based 3D/4D image viewer to easily and quickly view and evaluate images. This paper briefly describes the initial implementation of the web-based application.

**Keywords:** preclinical imaging; biomedical imaging; image data management; image data storage; PACs; medical imaging; multimodality imaging; cancer imaging; image data science

## 1. Introduction

Preclinical studies using animal models are an important bridge between in vitro experimentation and clinical trials commonly used to establish appropriate imaging protocols, understand biological principles of disease processes, evaluate disease responses to therapy, and translate results to clinical practice [1,2]. Medical imaging has emerged as an invaluable tool in both preclinical and clinical research for its capability to detect disease, monitor disease progression, and evaluate treatment response noninvasively and quantitatively [3,4]. Several imaging modalities have been developed over the last three decades [3–5]. These imaging modalities play an integral role in discovering basic pathological and biochemical mechanisms and in developing and evaluating novel diagnostic and theranostic approaches [6,7]. Imaging living subjects in vivo longitudinally expands conventional in vitro experiments and minimizes the use of animals for research [8–10]. Multimodality imaging is also commonly used to fully characterize the underlying anatomical, physiological, and functional processes by combining the information attainable with

different imaging modalities [11,12]. Software tools allow for the processing, co-registration, analysis, and visualization of images collected from different modalities [13].

A rapidly emerging literature illustrates the widespread application of multimodality imaging both in preclinical and clinical applications including cancer research, immunology, and infectious diseases [3–12,14–16]. Such widespread applications of imaging generate large volumes of highly detailed and heterogeneous image data. However, despite the availability of such a large amount of image datasets, there are still challenges in translating preclinical studies into clinical practice. This is not only due to common issues such as data incomparability, lack of standardization, and diverse experimental procedures, but also the lack of good data management tools and shared scientific practices that ensure data integrity, findability, accessibility, interoperability, and reusability according to FAIR guiding principles [17]. The new deep learning era also demands another pressing need for good data management tools and database infrastructures that provide access to organized datasets and facilitate reuse for various applications [18–22]. Such methods require large, curated datasets [23–25]. To address these impelling needs, several software solutions for image data storage and management have been developed [17,19,26–33]. While most products are specifically tailored toward clinical imaging, some of these systems have been adapted for use in preclinical research [30,31,33]. As, however, customization for preclinical applications is performed either using a top-down approach or as an extension to existing clinical tools [30,33], it is difficult to fully address the unmet needs in this area. This is also part of our experience in attempting to adapt ePAD, a quantitative imaging platform [34] developed by Rubine Lab at Stanford for preclinical image data management generated in our small animal facility.

Preclinical images are often stored in company-specific proprietary formats, so they must first be converted to more standardized formats such as DICOM or NIfTI for viewing. Furthermore, the types of associated metadata for preclinical imaging differ from those of clinical imaging [30]. Some of these programs include file conversion features to upload non-DICOM images but only provide limited information and capability to add specific information to each image data file. Detailed metadata recording of images and experimental details is important, as it provides information needed to properly interpret a given dataset. For example, a secondary image data analysis would not be adequate if information about the type, timing, dose, and route of administration of a specific contrast agent used are missing. An ideal platform will also store all associated raw data files to be able to reconstruct the images using different sets of parameters, which would eliminate other decentralized data archiving systems. It would also record all detailed metadata that can be pooled and applied for further analysis and reuse.

As we know, preclinical studies are performed to identify lead candidate drugs, methodologies, formulations, and other information needed for the ultimate design and input for clinical trials. This creates big volumes of data, much bigger than what may be needed for clinical studies. The data often goes through several steps of pre-processing to filter out selected relevant information for the translation to clinical trials. Hence, a data management tool based on the foundation of preclinical studies and built on a bottom-up approach may have better capacity, efficiency, accuracy, flexibility, and simplicity to address the unmet needs of data management in both preclinical and clinical studies. This was the main motivation that helped initiate this study. In this paper, we briefly describe the initial development of a new web-based server application for multimodal preclinical imaging data storage and management, which eventually might be developed into a full application for Biomedical image Registration, Analysis, and Translation (BiRAT).

## 2. Materials and Methods

### 2.1. Conceptual Design and Data Model

Small animal imaging using mouse models has now become a standard in preclinical and translational research [1]. As a result, there is big investment and establishment in shared core imaging facilities in almost all big academic and non-academic research

institutions. Such facilities are often equipped with various types of imaging modalities that facilitate biomedical research through imaging and generate a large volume of data. Hence, the base design and goal of this application is to develop tools that allow data capture and registration immediately after their creation to a centralized archiving system and following them during any subsequent processing and analysis steps.

The application uses a data-driven bottom-up approach to record each data and its relationship to specific experiments, studies, and projects. Figure 1 illustrates the conceptual design and data model of the application. The original unstructured data collected from various imaging modalities and related animal preparation experiments are annotated and stored in a structured hierarchical database. Data can be managed, accessed, and shared through either private or public web clients for better data security and accessibility. The application also includes a robust image viewer and modular processing tools for various computationally intensive and AI applications. A centralized distributed data storage is used for accessibility, scalability, security, and performance.

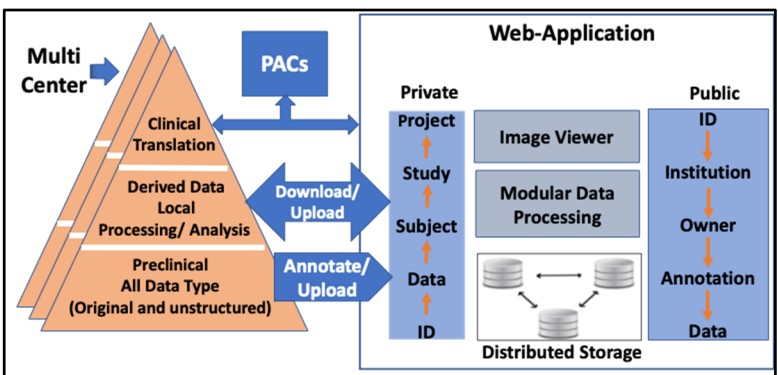

**Figure 1.** Conceptual design and data model for Biomedical image Registration, Analysis, and Translation (BiRAT). The model illustrates the workflow on how pieces of data elements are curated and added early immediately after its creation to the database system. Robust image viewer facilitates quick image preview and analysis.

### 2.2. Web-Server Application

The web application was developed using Django, a Python-based web-development framework [35]. Django was chosen due to its ease of use for rapid prototyping, scalability, and flexibility. In addition, it allows easier integration of any image processing and analysis tools that will be developed using Python libraries. The front-end user interface was written using HTML and JavaScript. The backend was built using Django with MariaDB, an SQL-based database management system. The application was containerized using Docker to ensure the isolation of the application development environment and streamline deployment [36]. The application was implemented using Python 3.7, Django 3.0, and Docker 20.10. Additional image reading and mathematical functionality were implemented using numpy 1.19 (general mathematical and scientific computing), pydicom 2.1 (DICOM file reading and writing), nibabel 3.2 (NIfTI file reading and writing), and Pillow 7.1 (general image reading and writing). The current implementation of the application is built using only Django, utilizing the Django templating languages to pass information from the application backend to front-end user interface. Future implementation will involve using the Django REST API for the back-end and Angular for the front-end development. This implementation will enable modularization, making it easier to change different aspects of the application as needed and aid in scaling up the application.

### 2.3. Image Viewer

The application includes an easily accessible image viewer that also supports common preclinical image types available on modern browsers. To minimize the extensive use of frameworks and libraries, and to make the application as lightweight as possible, the

viewer was designed using Cornerstone, a JavaScript library created for web-based medical image viewing applications [37]. Cornerstone reads binary data from a URL through a customizable image loader, then renders the image onto an HTML canvas. Cornerstone can separate large four-dimensional images into two-dimensional slices on the server or database, enabling the application to load each image individually only when needed by the user. This method is imperative for three-dimensional and four-dimensional images very common in preclinical imaging, which could take an extremely long time to load prior to being visible on the viewer. Cornerstone also provides a premade library and tools for reading DICOM images and performing basic navigation and analysis tasks. As a result, Cornerstone is still one of the most widely used frameworks for image viewing to date.

The customized viewer is based primarily on four JavaScript classes: the Container, the Border, the CSImage (or Cornerstone Image), and the Layer. The Container class is used to maintain properties related to the HTML division containing the CSImage. Most importantly, it defines how large the CSImage appears on the display. As multiple CSImages can be viewed simultaneously, there can be multiple Containers. Therefore, the Border class is used to define where a Container exists in relation to other classes. As multiple images can be viewed on top of one another, it is necessary for the CSImage to have the ability to contain not just information about a single image, but a list of images. Thus, the CSImage class maintains a list of images (often called Layers) as well as shared information that all Layers use. Lastly, the Layers class contains information about each individual image, such as a list of URLs that can be used to access binary data for each slice. One of the challenges of developing an image viewer for a wide variety of image formats is that of proprietary image formats, which Cornerstone does not support. We address this issue by exporting such files into Tagged Image File Format (TIFF) as most developers are more likely to support conversion to TIFF rather than DICOM.

### 2.4. Server Architecture and Storage

The back-end application is containerized using Docker [36], which allows for easy management of multiple instances if scaling is necessary due to increased user activity or for high availability. The back-end containers are placed behind Traefik [37], a load balancer to efficiently balance traffic between multiple servers on the network.

Imaging data are stored on multiple storage servers in a distributed manner. GlusterFS [8] is used as an abstraction layer on top of the filesystem of the physical disks, providing a transparent layer for replication and synchronization with automatic failover, providing high availability. GlusterFS is mounted on the individual back-end container instances, allowing direct access to datasets for the back-end. Furthermore, Samba shares can be used to mount project-specific folders to the site of data acquisition, allowing easy, secure, and immediate upload capability of project-specific files.

### 3. Results

Figure 2 outlines the current implementation structure of the application and how users can interact with it. After data are acquired, users can upload the data to the system via the user interface. These data, along with relevant metadata, are stored in the back-end. While manual data entry is the main subject of this paper, we are also looking to implement semi- or fully automatic data archiving methods to facilitate data recording and encourage users to use the platform. This would allow for the application to directly connect to each data acquisition instrument and automatically add images to the application without, or at least reduced, need for user interaction.

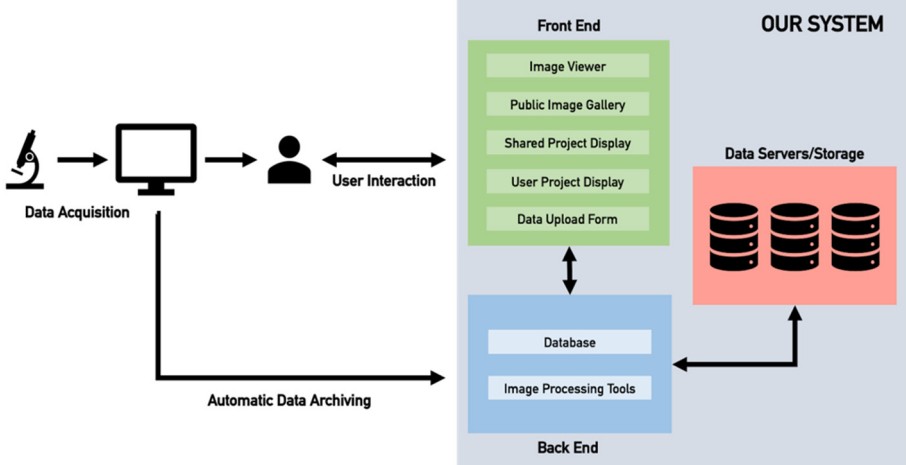

**Figure 2.** Current implementation of the structure and components of the web-based application designed on a data-driven bottom-up approach for efficient biomedical image data storage and management.

### 3.1. Data Storage Structure

One of the central goals of this project is to archive data that best reflects the actual experimentation workflow, as doing so would encourage integration into current practice. Therefore, special consideration was made when deciding how to structure data on the back-end. Figure 3 highlights selected tables from our data storage schema that are most relevant to the experimental workflow. As shown, data is organized hierarchically with projects, which can contain multiple studies, and studies, which can contain multiple image series. This system also allows users to manage research subjects (such as animal models and cell cultures) and link them to different studies, projects, and individual image series. An image series can contain any number of individual files to be viewed as a series of images in the image viewer. This method of maintaining image data is similar to the way images are stored using the Digital Imaging and Communications in Medicine (DICOM) standard, where a set of related 2D image slices can be viewed together as a 3D volume [34]. The image series structure is also useful for image data that is stored with one file containing raw image data and another header file containing image metadata.

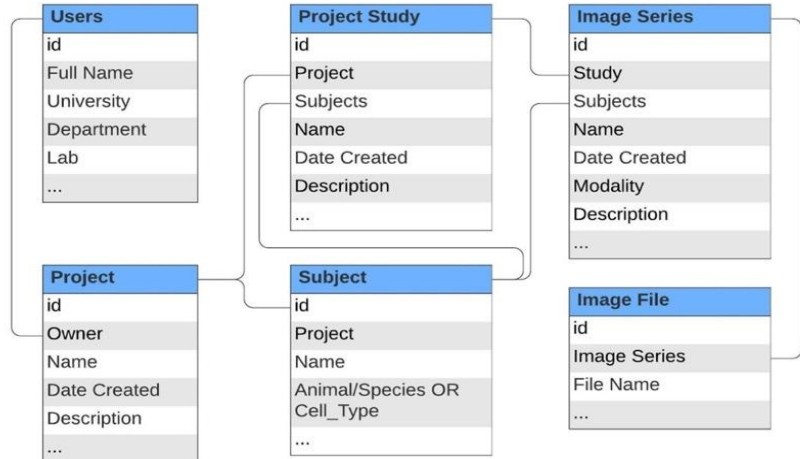

**Figure 3.** Simplified database schema of the application.

While a data structure can be interpreted and exploited in several different ways to best serve the individual user's needs, we consider this model to be a relatively standard way of breaking down different parts of the research workflow. Once a project and its

corresponding goals are determined, relevant experiments will be performed on a set of animal or cell subjects, which will generate images and other types of data. We expect an easy adaptation of the application to any project, as it does not require users to change how they structure projects to accommodate the usage of the platform. This is a specific unique feature of the application, which is not directly supported by other existing data management systems.

### 3.2. User Interface

Figure 4 demonstrates the "Study Manager" page of our application. This page allows users to view, upload, update, and delete image data belonging to a specific study. Users can use the icons provided to view the images in their browser via the image viewer. From this page, the user can also send individual image series to a public image registry, making the images available for viewing for all users of the site. Similar pages exist to allow users to manage data related to all their projects and all of those projects' studies. This structure allows for separate record-keeping of useful metadata recording of all projects, all studies, and individual image files as well.

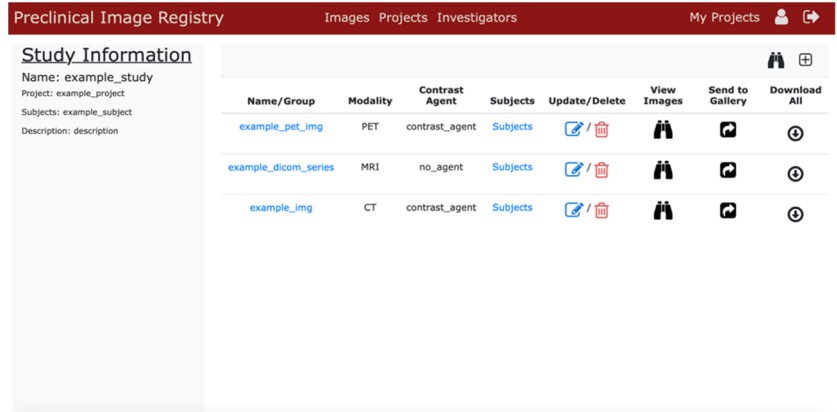

**Figure 4.** Screenshot of the "Study Manager" view.

### 3.3. Image Viewer

Figure 5 shows the image viewer while showing multiple mice PET images acquired from a small animal PET/CT scanner. This viewer can be used to visualize one or more selected images. The viewer features an easily navigable set of standard medical image viewing tools, such as annotation and segmentation tools. The viewer is capable of loading and overlaying multiple images in both 3D and 4D image formats, making it easy to view and compare multiple related images simultaneously.

The current version of the image viewer supports a number of features available on other common browser-based image viewers such as OHIF [38]. Most common medical image formats, such as TIFF, DICOM, and proprietary Siemens Inveon PET/CT, are supported. Some common image navigation and analysis tools are also implemented including navigation tools (panning, zooming, etc.), projection onto perpendicular axes for multi planar reconstruction (MPR) viewing, three-dimensional location synchronization (crosshairs), overlaying of two images, color schemes, length, and angle measurements, basic 2D segmentation, and region of interest tools. The unique feature of the current viewer is its capability to dynamically load images in real time. A preliminary test still unoptimized implementation of this feature was able to load 4D image dataset containing 3760 slides $512 \times 512$, 0.9 gigabytes in seconds. The viewer is also able to parse pages of both DICOM and TIFF images which, as defined by the user, can display basic information such as resolution, dimension, data of acquisition, etc., about the image displayed.

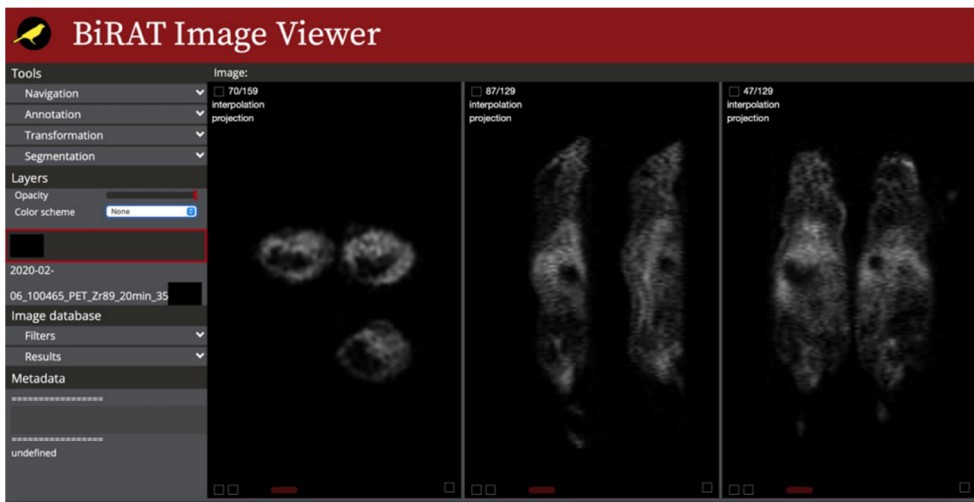

**Figure 5.** Screenshot of the application image viewer.

*3.4. Public Image Repository Portal*

The application provides easy access to send images to a public accessible image repository portal. Using this web portal, data sharing can be performed easily provided the receiver has authorized login credentials for the application. Users can choose to send their private images to this image gallery for public viewing. Images in the public portal can be searched using search parameters such as user info, date of acquisition, and by selected keywords. Data sharing is fully controlled and protected by credentials accessible to only authorized users. Figure 6 illustrates the image registry user interface.

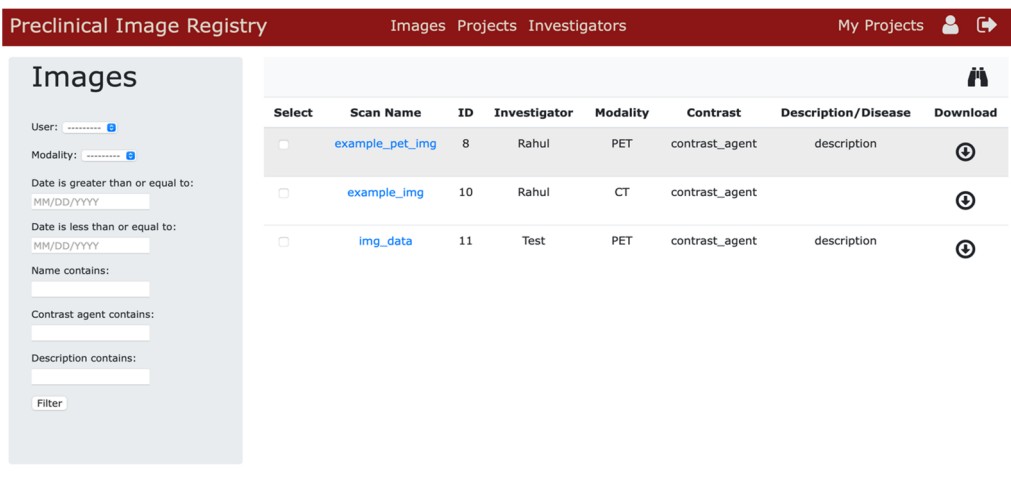

**Figure 6.** Screenshot of the public image registry or global image gallery.

## 4. Discussion

In this paper, we have presented the first steps towards building a comprehensive web-based biomedical image data management system primarily tailored for preclinical and translational imaging applications. The platform has been designed to address growing common issues in preclinical data storage and management while also fostering collaboration within the preclinical and translational imaging research community. Centralized core facilities housing several dedicated imaging modalities play an important role in providing access to both in vivo and ex vivo imaging for broad preclinical and clinical studies [39]. The challenge is that with increased use of such faculties, the need for robust data storage, management, and efficient analysis tools become critical while also limiting

the progress in basic science discovery and translation to clinic [19]. Additional issues related to proprietary file formats and non-standard practices require other dedicated data handling software tools specific for preclinical imaging, not yet fully developed [40]. As a result, most currently used image and analysis workflows are highly disjointed and cumbersome, often requiring many different machines, computers, and software. The inefficient processes affect all aspects of research and make it hard to collaborate [41]. It also limits the development of machine learning applications, as such methods depend on repositories of large amounts of data properly annotated and easily accessible [42], which is still very challenging to maintain.

Most of the current preclinical image data management systems use their best effort to utilize existing clinical tools to adapt for preclinical use [17,19,26–33]. XNAT-PIC [33] is a more recent development adapted as an extension to XNAT [43] a free open source for medical image repository. The major currently added feature of XNAT-PIC, however, is a tool that performs conversion from proprietary DICOM format from Bruker scanners to the DICOM format that XNAT supports. Future development to include data converters to support other preclinical modalities is also being worked out. Although using this approach leveraging existing data management infrastructure may provide a quick fix to some of the critical requirements for data management in preclinical research, it does not fully address the problem. Unlike clinical researchers, scientists using preclinical imaging deal with various complex types of data. There is often no bandwidth for the effort the user makes to upload and convert their data voluntarily or as added bonus for extra storage and repository purposes. Unless there is a direct relevance to their specific study that the user is compelled to use data management tools, most prefer to use other quick and easy local data storage solutions for the sake of saving time. This is the most difficult challenge to address in developing a robust and widely used preclinical data management system.

Small Animal Shanoir [30] is another cloud-based system developed to manage and process imaging datasets with more advanced features that allow management of metadata associated with each study beyond data storage and more efficient data sharing capability. The fact that this application is not freely available provides added issues, as it is not yet very popular, especially within the preclinical imaging facilities. Other recently developed commercial systems based on common preclinical workflow and that can also support multimodality data storage and management are SABER [19] and Flywheel [44]. By contrast, we are developing a new data management system specifically designed to directly support preclinical imaging workflow for all modalities and experimental setups. The goal is to develop a web-based application that can aid in improving data management at the individual lab level by changing the basic approach of data management system development strategy. A system using data driven bottom-up could make the application an integral part of the data acquisition system and potentially have wider use within the imaging research community.

## 5. Conclusions

The work presented in this paper highlights the key features of the initial development of the preclinical image data storage and management application. The application is being tested in the field while further refinement and development of the system are in progress. Further development is also being carried out utilizing modular implementation of the application to allow different parts of the system (front-end and back-end) to be handled and implemented independently. This will allow for greater flexibility and control throughout its full development.

**Author Contributions:** Conceptualization, R.P., R.M. and F.H.; methodology, R.P., R.M., K.S. and F.H.; software, R.P., R.M. and K.S.; validation, R.P., R.M. and E.W.; formal analysis, R.P., R.M. and E.W.; investigation, R.P., R.M., E.W. and F.H.; resources, F.H.; data curation, F.H.; writing—original draft preparation, R.P. and F.H.; writing—review and editing, R.P., F.H., K.S. and H.D.-L.; visualization, R.P.; supervision, F.H.; project administration, F.H.; funding acquisition, F.H. and H.D.-L. All authors have read and agreed to the published version of the manuscript.

**Funding:** This work was supported by the Canary Cancer Research Education Summer Program Grant (v R25-CA217729-03) funded by the National Cancer Institute and Stanford Center for Innovation in In vivo Imaging—small animal imaging facilities at Stanford. Daldrup-Link acknowledges support by NIH grant U24CA264298 from the NCI Co-Clinical Imaging Research Resource Program (CIRP) and the Stanford Cancer Institute Support Grant (5P30CA124435-10).

**Institutional Review Board Statement:** Not applicable for studies not involving humans or animals.

**Informed Consent Statement:** Not applicable for studies not involving humans.

**Data Availability Statement:** Not applicable.

**Acknowledgments:** We acknowledge the support of Steve Hisey and Max Meloche for all IT-related support and setting up the data server for the project.

**Conflicts of Interest:** The authors declare no conflict of interest.

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
