# Peer review of "Web-Based Application for Biomedical Image Registry, Analysis, and Translation (BiRAT)"

_tomography, doi:10.3390/tomography8030117_

Round 1

Reviewer 1 Report

The manuscript describes a comprehensive web-based biomedical imaging data management system that is useful to multimodal preclinical imaging research. The developed software platform allows capture and registration of the original unstructured data collected from various imaging modalities. Experiments are annotated and stored in a structured hierarchical database that allows subsequent processing and analysis. The application also includes a robust image viewer and processing tools for various applications. The viewer is capable of loading and overlaying multiple images at once in both 3D and 4D image formats, as sounds very convenient. The developed platform might be useful in artificial intelligence applications and for retrospective image analyses. The manuscript is well written; includes schematic presentation of the conceptual design and application components. Good solid work. I don’t have additional suggestions to authors for manuscript improvement.

Author Response

Thank you for constructive review, minor spell check has been performed. 

Reviewer 2 Report

The paper is very well written, clear to understand and well documented. The only question in my mind, at first, was on why limit the scope to preclinical studies but then it becomes clear that the objective is to support preclinical studies and not yet another generic clinical data management system. So I think this approach is very welcome in terms of supporting preclinical studies in a systematic way.

Author Response

Thank you for constructive review, minor spell check and style has been reviewed. 

Reviewer 3 Report

This paper described a web-based application for biomedical image management. It utilized several software frameworks, such as Django, Cornerstone, Docker, in developing the software. Overall, the paper is clear and concise. However, a few concerns may be addressed before the paper is being considered for publication.

  • In the Abstract, it mentioned that the artificial intelligence applications may be integrated into BiRAT. However, I didn’t see anywhere in the paper address that. Have the authors implemented this feature?
  • All figures in this paper are screenshots of BiRAT. Can the authors clarify how many features have been implemented? For example, what items were included in the drop-down menus? Can the authors provide a few example figures?
  • The author mentioned in the paper several times that the BiRAT is designed for handling large datasets. Has it been tested? How large the test date was?
  • Can the author provide some details about what the users are expected to do if they want to add the support for new format images?

Author Response

Thank you for constructive review, minor style and spell check have been reviewed. 

Please also see attached point by point response to comments and suggestions

Round 2

Reviewer 3 Report

Looks good!